# Biological Effects of Music Therapy in End-of-Life Care: A Narrative Review

**DOI:** 10.3390/medicina61091690

**Published:** 2025-09-18

**Authors:** Stefano Terzoni, Antonino De Vita, Paolo Ferrara, Francesco Sacchini, Giovanni Cangelosi, Stefano Mancin, Fabio Petrelli, Diego Lopane, Alessandra Milani, Mauro Parozzi, Maura Lusignani

**Affiliations:** 1Department of Biomedicine Science for Health, University of Milan, via Pascal 36, 20133 Milan, Italy; stefano.terzoni@unimi.it (S.T.); maura.lusignani@unimi.it (M.L.); 2IRCCS National Cancer Institute, via Venezia 14, 20133 Milan, Italy; 3ASST Santi Paolo e Carlo, Via Ovada 26, 20142 Milan, Italy; paolo.ferrara@asst-santipaolocarlo.it; 4Department of Nursing, Polytechnic University of Ancona, via Tronto 10, 60126 Ancona, Italy; francescosacchini@libero.it; 5Experimental Medicine and Public Health Unit, School of Pharmacy, University of Camerino, via Madonna delle Carceri 9, 62032 Camerino, Italy; fabio.petrelli@unicam.it; 6IRCCS Humanitas Research Hospital, via Manzoni 56-Rozzano, 20089 Milan, Italy; diego.lopane@hunimed.eu; 7IEO, European Institute of Oncology IRCCS, 20141 Milan, Italy; alessandra.milani@unimi.it; 8Department of Medicine and Surgery, University of Parma, via Gramsci 14, 43126 Parma, Italy

**Keywords:** end-of-life care, music therapy, biological effects, nursing care, narrative review

## Abstract

*Background and Objectives*: Music therapy has a long tradition in palliative care, and recent studies have investigated its Neuro-Psycho-Endocrine-Immunological (NPEI) effects in terminally ill patients. Despite numerous published articles, there is a lack of a compendium connecting the physiological basis of music therapy with the specific musical elements most effective in end-of-life settings. This narrative review aims to synthesize current evidence on the physiological mechanisms underlying responses to music, with a focus on terminal patients and implications for nursing practice. *Materials and Methods*: For quality and possible reproducibility, a narrative review was conducted in accordance with Scale for the Assessment of Narrative Review Articles (SANRA) guidelines. The review targeted articles from the past five years indexed in PubMed, CINAHL, Cochrane Library, Embase, Scopus, Web of Science, and PsycInfo, supplemented by additional relevant references identified through manual searching. The PICOS framework was performed to structure the search strategy and study selection, focusing on studies relevant to the biological effects of music therapy in end-of-life care and their practical implications for nursing care. *Results*: The neurophysiology of music perception in terminal patients is complex, involving a wide array of clinical and cultural factors. Key musical elements—such as rhythm, melody, harmony, tempo, and mode—can influence physiological and psycho-emotional responses. Music therapy interventions, when tailored to the individual’s preferences and cultural background, may modulate parameters like heart rate, blood pressure, stress hormone levels, and pain perception. Evidence supports the need for individualized approaches and highlights the NPEI rationale for integrating music therapy into end-of-life care. *Conclusions*: A deeper understanding of the scientific mechanisms discussed in this narrative review can enhance the effectiveness of music therapy interventions in end-of-life settings. Nursing practice can benefit by integrating evidence-based selection of musical pieces and personalizing interventions to the clinical and cultural profile of each patient. Further interdisciplinary research is needed to establish standardized criteria for music therapy in palliative care and to optimize outcomes for terminally ill patients.

## 1. Introduction

The utilization of music therapy in end-of-life care has increasingly been recognized for its potential to enhance patient well-being during profoundly challenging life phases. Terminally ill patients frequently experience increased anxiety, depression, and reduced quality of life. Recent literature underscores music therapy’s efficacy in alleviating these symptoms and promoting holistic patient wellness. For instance, systematic reviews indicated significant psychosocial improvements and enhanced psychological and physical outcomes among adult cancer patients following music therapy, highlighting clear connections between music interventions and emotional health at life’s end [1,2]. Statistical evidence reveals that approximately 68% of palliative care patients experience moderate to severe anxiety [3]. Music therapy offers a supportive intervention significantly reducing these distressing symptoms. Meta-analyses have demonstrated clinically relevant reductions in pain and anxiety, as well as enhanced quality of life in terminally ill patients receiving music therapy compared to standard care [4]. Other studies have highlighted the significant positive impact of music therapy on spiritual well-being among advanced cancer patients, addressing existential concerns common in terminal diagnoses [5]. Specific personalized interventions have proven particularly effective; randomized multicenter studies on biographical music therapy have reported enhanced emotional expression and greater clarity in patients’ life narratives [6]. These findings align with evidence emphasizing music therapy’s beneficial effects on patient-centered outcomes in palliative contexts [7]. The physiological rationale underpinning music therapy involves complex neuropsychological mechanisms. Psychological experiences evoked by music stimulate specific neurological structures, subsequently modulating endocrine secretion, which affects immune system responses—this interplay defines the Psycho-Neuro-Endocrine-Immunological (PNEI) framework [8]. Such physiological interactions are critical in understanding how music therapy effectively reduces stress, anxiety, and associated physiological symptoms such as increased heart rate and elevated cortisol levels [9]. Musicological components—rhythm, melody, harmony, and tempo—also profoundly influence therapeutic outcomes. Aligning music therapy choices with specific musicological attributes significantly affects patient responses [10]. This alignment ensures personalized interventions, maximizing therapeutic efficacy by addressing individual patient preferences and emotional states [11]. Additionally, music therapy has demonstrated unique benefits when compared to other therapeutic modalities. Some studies have described its superior effectiveness in fulfilling emotional and psychological needs compared to aromatherapy or massage therapy [12]. Others have corroborated these findings, highlighting music therapy’s efficacy in pain modulation in palliative care settings, reinforcing its vital role in comprehensive patient management [13]. The benefits of music therapy extend beyond patients, significantly impacting informal caregivers. Evidence confirms reduced caregiver stress and anxiety pre- and post-bereavement due to music interventions, supporting holistic care approaches encompassing the patient’s entire support system [14]. Improved quality of life among caregivers receiving music-based interventions has also been observed. Emerging research continues to highlight music therapy’s broader applicability in symptom management and quality-of-life improvements, further emphasizing its integral role in palliative care. Comprehensive analyses consistently reinforce music therapy’s broad therapeutic potential [15,16,17,18,19,20,21,22,23,24]. Accumulating evidence firmly establishes music therapy as a crucial, multifaceted approach to end-of-life care, significantly addressing anxiety, pain, spiritual well-being, and caregiver support. Integrating detailed neuropsychological and physiological insights into clinical practice ensures effective, personalized therapeutic interventions, providing terminally ill patients comfort, dignity, and peace during their final life stages.

### Aims and Research Questions

The primary objective of this narrative review is to critically examine the biological effects of music therapy in terminally ill patients, with particular emphasis on its impact on anxiety, pain modulation, and overall quality of life enhancement.

The secondary objective is to explore the practical implications of these findings for nursing view, specifically focusing on the development and implementation of personalized music therapy interventions tailored to optimize therapeutic efficacy for both terminal patients and their caregivers.

How does music therapy biologically influence anxiety, pain, and quality of life in terminally ill patients?

What are the practical strategies and clinical implications for personalizing music therapy interventions to maximize their effectiveness in end-of-life care settings for patients and caregivers?

## 2. Materials and Methods

### 2.1. Study Design

This narrative review was conducted to synthesize current evidence regarding the biological effects of music therapy in end-of-life patients, with particular attention to its physiological mechanisms and implications for nursing practice. To ensure quality and possible reproducibility, the review followed the Scale for the Assessment of Narrative Review Articles (SANRA) guidelines Appendix A [25].

To enhance the rigor and clarity of the literature selection process, the PICOS framework [26] was applied as follows:

P (Population): Terminally ill patients in palliative and end-of-life care settings;

I (Intervention): Any form of music therapy (active, receptive, individualized);

C (Comparison): Standard care, other non-pharmacological interventions, or none;

O (Outcome): Biological/physiological effects (e.g., stress reduction, pain modulation, neuroendocrine effects), psychosocial well-being, impact on caregivers;

S (Study design): All type of study designs (primarily, secondary or grey literature) relevant to search conducted.

### 2.2. Data Sources and Search Strategy

A search was conducted across the following international databases: PubMed, CINAHL, Cochrane Library, Embase, Scopus, Web of Science, and PsycInfo. The search was limited to articles focused on studies investigating the physiological mechanisms and biological outcomes of music therapy interventions in end-of-life care. Additional relevant sources were identified through manual searches of the university’s scientific library and by screening the reference lists of included studies. Articles older than five years were included only if cited in recent reviews or deemed essential for a comprehensive discussion. Only articles in English were considered. The search strategy was developed using Boolean operators, wildcards, advanced search features, and thesaurus terms specific to each database. The keywords adopted were: “Terminally ill patients”, “Palliative care”, “End-of-life care”, “Music therapy”, “Active music therapy”, “Receptive music therapy”, “Individualized music therapy”, “Standard care”, “Non-pharmacological interventions”, “Biological effects”, “Physiological effects”, “Stress reduction”, “Pain modulation”, “Neuroendocrine effects”, “Psychosocial well-being”, “Caregivers”, “Anxiety”, “Quality of life”, “Music interventions”, “Therapeutic strategies”.

### 2.3. Inclusion and Exclusion Criteria

Studies were included if they

Explored the biological or physiological effects of music therapy in patients at the end of life;Provided practical indications for designing music therapy interventions;Met the population, intervention, comparison, and outcome criteria defined by the PICOS framework.

Papers that did not include the stated inclusion criteria were excluded.

## 3. Results

The results of this narrative review are presented according to a thematic structure that reflects the main domains of application and mechanisms of action of music therapy in end-of-life care. First, musicological aspects relevant to clinical practice and the importance of individual patient preferences in tailoring interventions are described. This is followed by an in-depth analysis of rhythmic, harmonic, and temporal parameters, as well as the influence of the patient’s musical training on therapeutic response. The review then addresses the neuroanatomical and physiological mechanisms involved in musical perception and the effects of music therapy on key physiological indicators. Finally, NPEI effects in cancer patients are discussed, with a focus on clinical implications and future perspectives.

### 3.1. Musicological Aspects of Clinical Relevance

Musicology identifies essential components such as rhythm, melody, harmony, and tempo, which significantly influence physiological and psycho-emotional responses in patients. Common music therapy interventions include singing, songwriting, lyric analysis, musical improvisation, imagery association, reminiscence, and relaxation techniques [2,27]. Music therapists continuously adapt interventions based on patient preferences and therapeutic needs, taking into account the clinical setting and the goals of the session [2,27]. For instance, in oncology or geriatrics, certain interventions may be favored due to patients’ cognitive or emotional states (Figure 1).

### 3.2. Importance of Individual Preferences

Selection of music should not be arbitrary; patient preferences, musical habits, and skills must guide the therapeutic choice to maximize engagement and efficacy. The collection of this information typically involves dedicated interviews or assessment tools aimed at understanding both musical taste and listening habits. Active versus receptive music therapy plays a distinct role, activating different brain areas involved in pleasure and emotional processing, as evidenced by EEG studies showing synchronized brain activity during music listening sessions [28,29]. This personalized approach also helps to foster a stronger therapeutic alliance and patient satisfaction (Figure 2).

### 3.3. Rhythmic and Temporal Considerations

Rhythm (even vs. odd) and tempo (beats per minute, bpm) significantly influence cortical attention and relaxation. Even rhythms typically provide a sense of completeness, whereas odd rhythms require greater cortical attention and may be stimulating for certain individuals. Tempos between 60–80 bpm are associated with relaxation, reducing sympathetic nervous system activity [30]. Elements like groove and swing rhythms also modulate emotional and physiological responses [30]. Clinicians may select rhythmic structures based on the patient’s clinical needs, such as promoting relaxation versus energization (Figure 3).

### 3.4. Harmonic Structure and Emotional Impact

Harmony and melodic intervals influence emotional and physiological responses. Major chords often evoke happiness, while minor chords elicit sadness or introspection. Modes such as Phrygian evoke distinct atmospheres and must consider cultural familiarity to ensure therapeutic efficacy. The intensity and dynamic range of music further modulate emotional experiences [30]. For example, classical Western music often uses major-minor contrasts, while some patients may respond better to modes and harmonies rooted in their cultural heritage (Figure 4).

### 3.5. Influence of Musical Training on Therapeutic Response

The listener’s musical training significantly impacts therapeutic effectiveness. Trained individuals show enhanced brain activity in areas responsible for analytical listening and emotional processing, enabling deeper therapeutic experiences and potentially greater therapeutic outcomes [29]. For instance, patients with musical background may benefit more from improvisational or analytical listening exercises, while untrained listeners may prefer simpler, familiar melodies. Assessing musical training can help tailor interventions to each patient’s capacities (Figure 5).

### 3.6. Neuroanatomy of Musical Perception

Music activates multiple cortical and subcortical brain regions, including the auditory cortex, amygdala, orbitofrontal cortex, and limbic structures involved in emotional regulation [31]. The physical sensation of music vibrations also contributes to therapeutic effects by activating additional sensory pathways [31]. Studies using techniques such as fMRI and EEG have mapped the activation patterns associated with both passive listening and active music-making, reinforcing the biological plausibility of music therapy’s benefits (Figure 6).

### 3.7. Physiological Effects of Music Therapy

Music therapy significantly reduces physiological stress markers, including heart rate, respiratory rate, blood pressure, and cortisol. These effects are mediated by decreased sympathetic nervous system activity and enhanced endorphin release, resulting in improved pain management and overall well-being [32,33,34,35]. For example, several studies have shown reductions in heart rate of up to 10–15% during music interventions, with corresponding decreases in reported pain and anxiety. These physiological changes contribute to improved quality of life in palliative care settings (Figure 7).

### 3.8. Psychoneuroimmunological Effects in Cancer Patients

Music therapy reduces stress hormone levels, particularly catecholamines, implicated in cancer progression and neo-angiogenesis. Reduction in these hormones may influence tumor biology by decreasing metastatic potential and angiogenesis, providing a biological rationale for integrating music therapy into cancer care [30,34,37,38]. Studies suggest that these effects may be more pronounced in patients who report high baseline stress or anxiety, but individual responses vary due to personal experiences and cultural backgrounds, necessitating personalized music therapy interventions [39]. Incorporating music therapy into oncology protocols can therefore be considered not only for symptom relief but also as part of a holistic strategy to support patients’ overall health (Figure 8).

## 4. Discussion

This narrative review highlights the multifaceted role of music therapy in end-of-life care, synthesizing recent evidence on musicological, neurophysiological, and NPEI mechanisms. The literature confirms that music therapy is a valuable adjunct in palliative contexts, offering clinical benefits that range from the modulation of psycho-emotional states to measurable physiological effects [6,13,41]. A central finding is the importance of musicological parameters—such as rhythm, melody, harmony, and tempo—in shaping therapeutic outcomes. Interventions must be tailored to individual preferences, habits, and musical training to maximize effectiveness [2,27,29]. Recent randomized trials and systematic reviews have shown how both active and receptive modalities engage distinct brain circuits and emotional processing, with biographical and legacy-based approaches being especially meaningful for patients and families, supporting emotional connection and closure [6,42]. The therapeutic efficacy of music is further refined by choices around rhythm (even or odd), tempo (with relaxing tempos typically between 60–80 bpm), harmony (major versus minor chords), and culturally familiar modes [2,30]. These factors influence relaxation and emotional responses and must be selected based on clinical goals and the patient’s cultural background [39]. Musical background has been found to enhance the therapeutic impact, as trained individuals display stronger activation in brain regions responsible for analytical listening and emotional processing, and assessing this variable is recommended in planning interventions [29,43,44]. Music listening activates a distributed network of cortical and subcortical structures involved in emotion and memory [31], and music therapy has demonstrated reductions in stress markers—heart rate, blood pressure, and cortisol—with positive effects on pain, anxiety, sleep, and well-being, as confirmed in meta-analyses and randomized controlled trials [3,9,13]. EEG studies have shown that group music therapy sessions can synchronize cortical activity and promote shared therapeutic experiences [28,45]. Reducing catecholamine levels through music therapy may help modulate tumor growth and angiogenesis in palliative oncology [37,46], but outcome variability underscores the need for individualized interventions based on biological and cultural factors [39]. Despite promising evidence, methodological heterogeneity persists, and recent reviews recommend more standardized protocols and robust outcome measures in both research and clinical settings [13,47]. Key practical recommendations include personalizing music therapy to patient preferences, training, and culture, using musicological criteria aligned with therapeutic objectives, considering group music therapy for shared benefit, promoting multidisciplinary teamwork among music therapists, clinicians, and nurses, and incorporating music therapy education into palliative care training programs [48,49,50]. While the literature largely supports the beneficial effects of music therapy in palliative care, it is important to also consider studies reporting limited or no significant effects, in order to provide a balanced perspective in the discussion [51,52].

### 4.1. Implications for Clinical Practice

Music therapy in these contexts have to be provided by qualified personnel that must be suitably trained in the biological effects of music in terms of PNEI and about the link between music characteristics and mechanisms capable of leading to clinically measurable effects. Like in other clinical settings [53,54] the application of some music therapy interventions could be directly managed by specifically trained nurses aiming the music therapy experience at treating pain [55] anxiety, depression and improving quality of life [54]. Integrating music therapy may enrich traditional approaches by supporting complex aspects such as pain, nutrition, and pharmacological management in palliative care, cancer and chronic care in general [56,57,58,59,60,61,62,63,64]. Nursing practice should combine information about the patient, their habits, their clinical path and their own musical taste with evidence about the biological and psychological effects of music therapy in order to guide song selection and try to optimize the desired effects, customizing the patient care. In addition, nurses should be trained in the use of specific validated tools to assess the patient experience and the biological and psychological effects of interventions in order to increase the quality of care in end-of-life settings. Partnering with music therapists, nurses can support a holistic and collaborative approach to end-of-life care and even change the dynamics within health systems [64]. To further support the clinical application of music therapy, it is advisable that nurses and clinicians refer to standardized music selection criteria or validated operational protocols, in order to guide the choice of pieces more systematically and optimize the desired effects, while still maintaining personalization according to the patient’s characteristics and preferences [48,64,65,66,67].

### 4.2. Limits

This study is a narrative review, a methodological approach that, while allowing a broad and integrative overview of the existing literature, presents some internal limitations. In particular, the selection of studies does not follow strict systematic criteria, and quantitative comparability among the reported results may be limited. Consequently, the synthesized data should be interpreted with caution, taking into account methodological and contextual differences among the included studies. Furthermore, the variability in intervention protocols, assessment tools, and the different clinical contexts considered makes it difficult to generalize the results and draw unequivocal conclusions. It is important to emphasize that the narrative nature of this study was chosen to provide a broad and integrative overview, rather than to produce consolidated quantitative estimates. Therefore, the results should be considered as preliminary indications, requiring further studies with more rigorous methodological designs to confirm and deepen the evidence presented.

## 5. Conclusions

Listening to music is a complex phenomenon, able to activate rational brain areas and other deeper ones, linked to memory and emotional processing. In situations such as terminality, it is precisely these characteristics that make music an extremely powerful tool to reach and process emotions of great intensity, sometimes long-standing and difficult to reach with more rational paths. This complexity is reflected in the need, by those who apply the therapeutic intervention, to consider many parameters and characteristics of the patient and of the pieces. This knowledge forms the basis for the selection of songs to be used for music therapy in the nursing care of end-of-life individuals. The literature appears to be lacking in work that provides a compendium of evidence regarding practical criteria for choosing music according to the PNEI considerations presented in this article; rigorous RCTs should therefore focus on these aspects of music therapy.

## Figures and Tables

**Figure 1 medicina-61-01690-f001:**
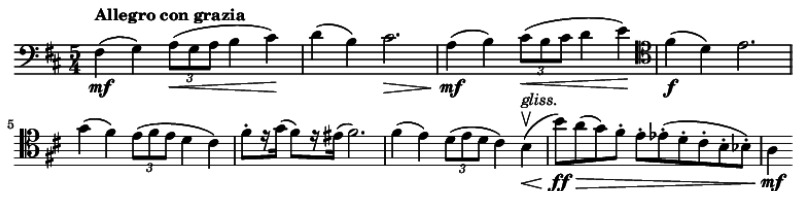
Opening theme of the 2nd movement, Tchaikovsky op 74 n.6. Note: used in the context of music therapy to support relaxation and reduce stress.

**Figure 2 medicina-61-01690-f002:**
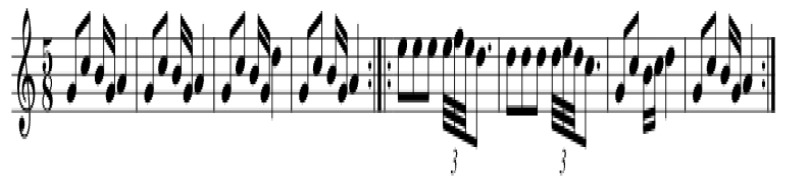
Tsakonikos (Greek dance) in 5/8. Note: music intervention aimed at promoting relaxation and well-being.

**Figure 3 medicina-61-01690-f003:**
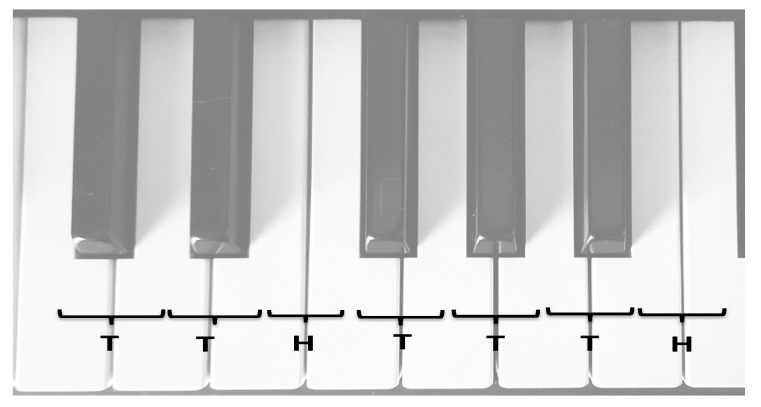
Major diatonic scale. T = Tone; H = Halftone. Note: shown as an example of a musical element used to support relaxation and emotional well-being in therapeutic contexts.

**Figure 4 medicina-61-01690-f004:**
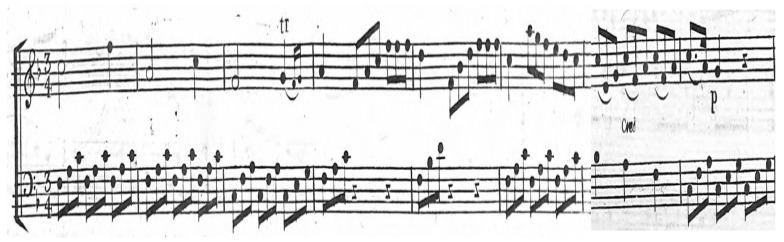
Example of voicing (Clementi M. Sonatina in F Major). Note: demonstrating a musical feature that may foster reflection and emotional support.

**Figure 5 medicina-61-01690-f005:**
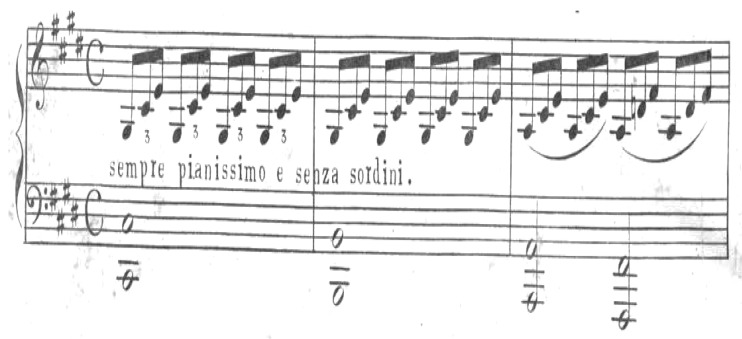
Sonata Op. 27 n.2, first movement, first three bars. Note: included to illustrate a musical element that may support reflection, emotional comfort, and general well-being in a therapeutic context.

**Figure 6 medicina-61-01690-f006:**
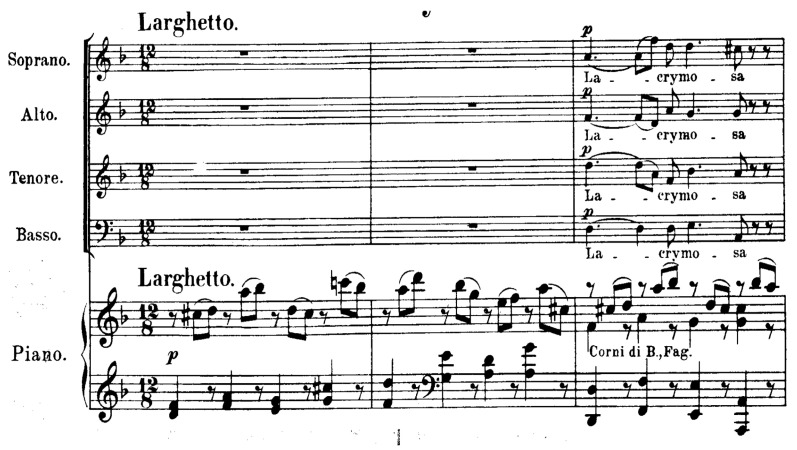
Lacrymosa (from the Requiem by W.A. Mozart), first three bars. Note: resented to evoke emotional resonance and moments of personal reflection in a therapeutic context.

**Figure 7 medicina-61-01690-f007:**
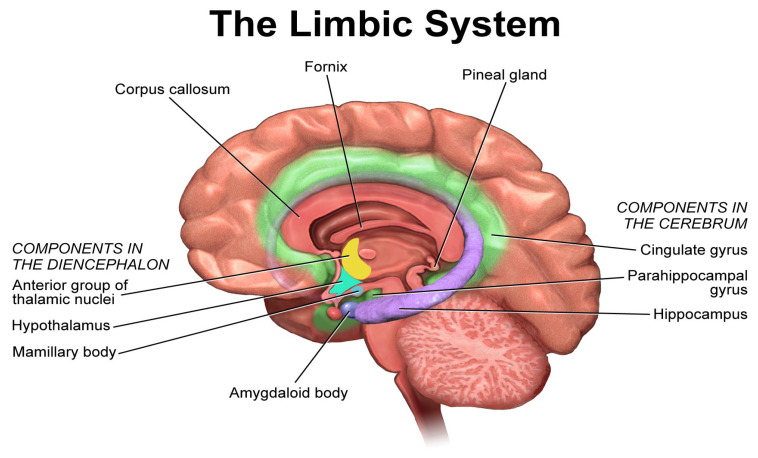
The Limbic System. (Blausen.com staff, 2014 [36]). (Source: Blausen.com staff (2014).”Medical gallery of Blausen Medical 2014”. WikiJournal of Medicine 1 (2). D0: 10. 15347 /wim/2014.010.ISSN2002-4436).

**Figure 8 medicina-61-01690-f008:**
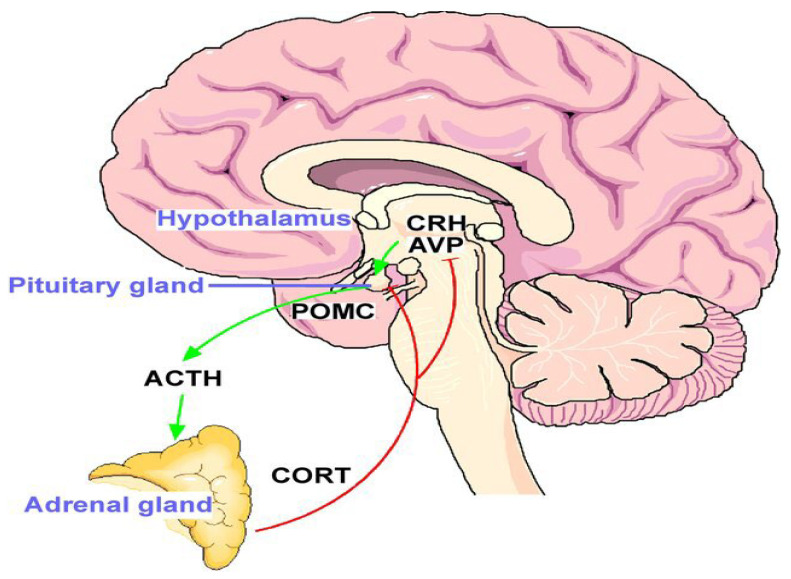
HPA Axis (Murgatroyd C and Spengler D, 2011 [40]). (Source: Murgatroyd, C.; Spengler, D. Epigenetics of early child development. Front Psychiatry. 2011, Apr 18;2:16. doi: 10.3389/fpsyt. 2011.00016.).

## Data Availability

The data supporting this research are available upon request from the corresponding author for data protection reasons.

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
