# Peer review of "Biological Effects of Music Therapy in End-of-Life Care: A Narrative Review"

_medicina, 2025, doi:10.3390/medicina61091690_

Round 1
Reviewer 1 Report
Comments and Suggestions for Authors
Dear Author(s),
I have carefully reviewed your article. The study examines the biological effects of music therapy in terminally ill patients from a comprehensive and interdisciplinary perspective and has the potential to make a significant contribution to the literature. However, to strengthen the scientific validity and clinical applicability of the study, I recommend the following revisions:
Methodology details: Data selection, inclusion/exclusion process, and the distribution of study types should be clearly presented (PRISMA diagram is recommended).
Strengthening quantitative findings: The magnitude and statistical significance of the obtained effects should be presented in tables or graphs.
Citation organization: Repeated citations and formatting errors should be corrected.
Balancing negative findings: Studies with ineffective or limited effects from the literature should also be included in the discussion.
Figures and music examples: Their relevance to the clinical context should be explained, and additional information to the reader.
Clinical practice recommendations: Standardized music selection criteria or protocol examples should be included for nurses and clinicians.
I believe that with these corrections, both the academic and clinical value of your article will increase.
Best regards
Author Response
Dear Peer,
thank you agein for your time and efforts. Please considerer suitable this second version of the manuscript after your relevant comments.
The authors

Reviewer 2 Report
Comments and Suggestions for Authors
Congratulations on the article!
This paper gives a clear overview of how music therapy can help people at the end of life, looking at both the science behind it and how it can be used in nursing care. It explains the different ways music can affect the brain, emotions, and body, and stresses the importance of choosing music based on each patient’s tastes, culture, and background. One weakness is that the authors describe the benefits in detail but do not provide much complex data or statistical analysis to prove cause and effect. It could also benefit from including more about how to measure results and whether this approach is cost-effective. Overall, it is a thoughtful and helpful review that can help guide better, more personalized care for patients at the end of life.
Author Response

(The authors gave the same response as above.)

Reviewer 3 Report
Comments and Suggestions for Authors
Overall, really great manuscript and I appreciated the topic of this work. Two things I wanted to note:
1. There was no justification or reasoning given for why you limited the articles to those no more than 5 years old. It would be great to have an understanding of why the decision was made.
2. In the results section, there was no mention of how many articles were included in the review and the number of articles that were relevant for each theme in your results. This information would be great to have just to give an idea of how much (or how little) research has been done exploring these topics.
Author Response

(The authors gave the same response as above.)

Round 2
Reviewer 1 Report
Comments and Suggestions for Authors
Dear Authors,
Thank you for submitting the revised version of your manuscript, “Biological Effects of Music Therapy in End-of-Life Care: A Narrative Review,” to Medicina. After carefully considering your responses and the revised files, I reviewed the manuscript based on the reviewers' comments.
Methodology: You appropriately adopted the SANRA checklist and provided the scoring in the Supplementary Materials. Furthermore, the use of the PICOS framework strengthened the reproducibility and transparency of the review process.
Literature Update: The revised manuscript now includes more current references (2023-2025), addressing reviewers' concerns about outdated literature. Meta-analyses and RCTs published in the last two years have been appropriately integrated.
Content and Depth: The revision successfully expands the scope of musicological parameters (rhythm, harmony, tempo, cultural context), neuro-psycho-endocrine-immunological mechanisms, and the role of patient preferences. These additions are in direct response to comments requesting greater depth and clarity.
Limitations: The limitations of the narrative approach are explicitly acknowledged, and the authors clearly address the lack of generalizability and the need for standardized protocols.
Practical Implications: Implications for clinical nursing practice, including validated tools, individualized interventions, and interdisciplinary collaboration, are clearly explained.
The revised article has been significantly improved and now meets the standards for publication in Medicina.
Best regards